# Sexuality, Intimacy, and Reproductive Health after Spinal Cord Injury

**DOI:** 10.3390/jpm12121985

**Published:** 2022-12-01

**Authors:** John Zizzo, David R. Gater, Sigmund Hough, Emad Ibrahim

**Affiliations:** 1Desai Sethi Urology Institute, Miller School of Medicine, University of Miami, Miami, FL 33136, USA; 2Department of Physical Medicine & Rehabilitation, Miller School of Medicine, University of Miami, Miami, FL 33136, USA; 3Department of Psychiatry, Harvard Medical School, Harvard University, Boston, MA 02215, USA; 4Department of Psychiatry, Boston University School of Medicine, Boston University, Boston, MA 02118, USA; 5The Miami Project to Cure Paralysis, Miller School of Medicine, University of Miami, Miami, FL 33136, USA

**Keywords:** spinal cord injury, sexuality, fertility, intimacy, ejaculation, relationship, reproductive health, erectile dysfunction, communication, pregnancy, semen quality

## Abstract

Spinal cord injury (SCI) is a life-altering event often accompanied by a host of anxiety-provoking questions and concerns in the minds of affected individuals. Questions regarding the ability to resume sexual activity, partner’s satisfaction as well as the ability to have biological children are just a few of the unknowns facing patients following the devastating reality that is SCI. As a result of advances in SCI research over the last few decades, providers now have the knowledge and tools to address many of these concerns in an evidence-based and patient-centered approach. SCI can impair multiple components involved in sexual function, including libido, achieving and maintaining an erection, ejaculation, and orgasm. Many safe and effective fertility treatments are available to couples affected by SCI. Finally, learning to redefine one’s self-image, reinforce confidence and self-esteem, and feel comfortable communicating are equally as important as understanding functionality in regaining quality of life after SCI. Thus, this review aims to highlight the current state of SCI research relating to sexual function, reproductive health, and the search for meaning.

## 1. Introduction/Epidemiology

Spinal cord injury (SCI) frequently occurs in men during the years of their reproductive health peak when they may desire to start a family and have children. In the United States, approximately 78% of new SCI cases are males, with the average age at injury being 43 years [1]. Similar statistics have been reported in other countries and regions [2,3,4,5,6,7]. The most common causes of SCI are motor vehicle accidents, falls, violence (including gunshot wounds), and sports-related injuries [1].

Following SCI, men face numerous complications, including erectile dysfunction and infertility. It is estimated that 90% of men with SCI cannot father a child naturally [8]. However, women with SCI can conceive and deliver children with nearly the same success rate as the general population [9]. When surveyed, patients affected by SCI ranked sexual and reproductive functions as major factors affecting their quality of life [10]. A large percentage of men with SCI suffer from erectile dysfunction (75%) [11], ejaculatory problems (95%) [12], and poor semen quality, necessitating medical assistance to have children [13]. Non-traumatic damage to the spinal cord such as disc herniation may have similar effects on sexual dysfunction.

## 2. Sexual Health after SCI

### 2.1. Erectile Dysfunction

#### 2.1.1. Erection Physiology

Reflex Erections are mediated by the sacral reflex arc, requiring intact S2–S4 nerve roots. They are produced by direct stimulation of the penis and occur independently of erotic stimuli. Reflex erections are generally preserved with injury above L2. On the other hand, psychogenic erections are mediated by the sympathetic nervous system and require intact thoracolumbar nerve roots. They are produced by visual or mental stimuli and occur independently of direct penile stimulation. Psychogenic erections are often disrupted due to injury to the thoracic and cervical spinal cord [14,15,16,17].

Cyclic guanosine monophosphate (cGMP) is responsible for the vascular changes in the corpora cavernosa that result in an erection. cGMP is hydrolyzed by phosphodiesterase-5 (PDE-5) to GMP, resulting in loss of penile tumescence. This process is initiated by endogenous nitric oxide (NO) activation of guanylate cyclase, resulting in the increased conversion of guanosine triphosphate (GTP) to cGMP. The inhibition of PDE-5 results in the maintenance of high levels of cGMP [18,19].

#### 2.1.2. Diagnosis

Diagnosing sexual dysfunction in males requires a thorough medical, sexual, and psychological history. Examination often includes the use of validated questionnaires, such as the International Index of Erectile Function (IIEF), which provides a reliable and standardized measure of erectile dysfunction. Hormonal profiles and a penile Doppler ultrasound to assess penile vascular flow are also utilized [20,21,22,23]. Assessment of male and female sexual dysfunction after SCI can be performed using the International SCI Male Sexual Function Basic Data Set Version 2.0 and the International SCI Female Sexual and Reproductive Function Basic Data Set Version 2.0 [24].

#### 2.1.3. Treatment

Management of erectile dysfunction in men with SCI follows the same standard of care protocols used in the general population. Early reports described erectile dysfunction in men with neurogenic complications, including SCI [25]. These included trials of FDA-approved oral phosphodiesterase type 5 inhibitors (PDE5i) such as Sildenafil, Tadalafil, Vardenafil, and Avanafil [26,27]. Multiple studies on men with SCI highlighted the safety and effectiveness of oral PDE5i in managing erectile dysfunction in this population [28,29,30,31,32,33,34,35]. Alternative modalities can also be utilized in men with SCI, including vacuum erection devices (VED) and penile constriction bands; direct instructions should be given to remove these bands immediately after sexual activities [36,37,38]. Few reports showed that men with SCI do not prefer the use of intra-urethral alprostadil as well as lower absorption due to the unhealthy urethral mucosa [39]. In cases refractory to PDE5i therapy, intra-cavernosal injections have shown promising results [40]. Finally, in severe cases, surgical placement of a penile prosthesis can be considered [13,41,42]. The decision between a malleable and inflatable prosthesis is individualized based on patient preference. Malleable prostheses are sometimes preferred by men with tetraplegia due to impaired hand dexterity, making activation and deactivation of the device challenging [43].

### 2.2. Anejaculation

#### 2.2.1. Ejaculation Physiology

Normal ejaculation requires the coordination of an intact ejaculatory reflex as well as parasympathetic, sympathetic, and somatic neural components. Ejaculation consists of two phases: (I) Emission, the deposition of seminal fluids and sperm in the posterior urethra, is mediated by the sympathetic nervous system, resulting in the contraction of the seminal vesicles and prostate. (II) Expulsion, the propulsion of ejaculate out of the urethra, is initiated by bladder neck closure to prevent retrograde ejaculation combined with contractions of the pelvic floor and bulbocavernosus muscles followed by relaxation of the external urinary sphincter. These actions are mediated by the sympathetic, parasympathetic, and somatic nervous systems. The final step involves the spurting of semen from the urethral orifice [44].

A study in 2016 provided evidence of a spinal generator of ejaculation (SGE) in humans. The organization and sexual dimorphism of human spinal galaninergic neurons were noted to be similar to those found in rats. It has been hypothesized that these neurons play a key role in ejaculation [45]. The SGE is located in the mid-lumbar spinal cord segments and is responsible for orchestrating the activity of thoracolumbar sympathetic, sacral parasympathetic, and somatic nuclei commanding anatomical structures participating in ejaculation [46].

#### 2.2.2. Treatment

As mentioned previously, most men with SCI cannot ejaculate via sexual activity or masturbation [47]. Fortunately, various methods of medically assisted ejaculation are available to overcome anejaculation in men with SCI, including penile vibratory stimulation (PVS), electroejaculation (EEJ), prostate massage, and surgical sperm retrieval (Figure 1).

### 2.3. Penile Vibratory Stimulation (PVS)

PVS is the first-line treatment for anejaculation in men with SCI [48,49,50]. During PVS, a medically approved high-amplitude vibrator is placed on the dorsum or frenulum of the glans penis [51]. The vibrator induces mechanical stimulation, thereby recruiting an ejaculatory reflex resulting in ejaculation [52]. This method has proven highly successful in men with SCI whose level of injury is T10 or rostral (88%) and slightly successful in caudal injuries (15%) [53]. If a patient is unable to ejaculate with a high-amplitude vibrator, two vibrators can be applied to the dorsum and frenulum of the penis [54]. Abdominal electrical stimulation in combination with PVS has been reported as a salvage method for PVS failure [55,56].

In men with SCI, extra precautions are needed in any assisted ejaculation procedure if the level of injury is T6 or rostral due to the risk of inducing autonomic dysreflexia [57]. Autonomic dysreflexia is a potentially life-threatening complication caused by stimulation below the level of injury, resulting in an uninhibited sympathetic reflex. Symptoms may include hypertension, bradycardia or tachycardia, sweating, chills, and a pounding headache. In severe cases, this can lead to dangerously high blood pressure levels resulting in a stroke, seizure, or even death. Autonomic dysreflexia symptoms can be managed or prevented by administering antihypertensive medications such as oral nifedipine prior to stimulation [58].

#### 2.3.1. Electroejaculation (EEJ)

The EEJ method was first developed in Australia in the 1930s for its use in veterinary medicine [59]. It was then modified and subsequently received FDA approval for human use in the 1980s [60,61]. Patients who fail PVS are referred for EEJ, which is performed with the patient in the lateral decubitus position. The rectum is visually examined, followed by the gentle insertion of a probe. Electrodes on the probe are oriented anteriorly toward the prostate and seminal vesicles. Electric current is gradually delivered in a stepwise manner, allowing the probe to stimulate nerves involved in semen emission [62]. Due to the elevated risk of retrograde ejaculation, bladder preparation is recommended by installing sperm wash medium after emptying [63].

#### 2.3.2. Prostate Massage

Prostate massage is an alternative method of obtaining sperm from anejaculatory men with SCI. However, there is a lack of consensus regarding its algorithmic placement and role, given the wide array of treatments available. Studies have successfully obtained sperm via prostate massage in men with SCI. However, results are inconsistent, with one report obtaining sperm from only 22 of 69 men with SCI (32%). Thus, prostate massage may be a reasonable step prior to surgical sperm retrieval in cases refractory to PVS and when EEJ is either unavailable or requires general anesthesia (GA), as this adds considerable cost to the EEJ procedure. GA is required in only a minority of SCI patients who retain pelvic sensation and who may therefore experience pain during EEJ [64,65,66].

#### 2.3.3. Surgical Sperm Retrieval

Sperm aspiration from the testis or epididymis via puncture or microsurgery can be performed to surgically retrieve sperm from reproductive tissues. Multiple techniques are available, including testicular sperm extraction (TESE), testicular sperm aspiration (TESA), microsurgical testicular sperm extraction (m-TESE), microsurgical epididymal sperm aspiration (MESA), percutaneous epididymal sperm aspiration (PESA), and aspiration of sperm from the vas deferens [67,68]. In a Japanese single-center study, TESE was achieved in 80.7% of patients with SCI-induced ejaculatory dysfunction [69]. Similarly, Raviv et al. found a sperm retrieval rate of 89.6% using TESE/TESA in men with C2-L2 SCI refractory to PVS or EEJ [70]. While promising results have been shown, surgical sperm retrieval yields a small seminal volume and fewer spermatozoa compared to ejaculate. Thus, these methods are typically reserved for cases refractory to both PVS and EEJ, as well as those involving obstructive azoospermia [71].

### 2.4. Semen Quality/Motility Barriers

Numerous studies have reported abnormal semen quality in men with SCI [8,72,73,74]. In particular, the macroscopic appearance is often abnormal, with 27% of men producing brown-colored semen [75]. Microscopically, several abnormalities have been noted, such as numerous white blood cells, as well as other debris. Additionally, sperm motility and viability are typically below the standard reference ranges. However, most studies show that men with SCI produce a normal sperm count [13].

Earlier studies attempting to elucidate the underlying cause for poor sperm quality in men following SCI focused on complications such as the presence of elevated scrotal temperature (attributed to sitting in a wheelchair), anejaculation, neurogenic bladder, and the use of assisted ejaculation techniques that are less efficient than natural ejaculation. However, scrotal temperature was found to be similar in injured and non-injured men [76], frequent ejaculation did not alter the semen profile [77,78,79], sperm motility failed to normalize with proper bladder management [80], and no major differences were noted between various methods of assisted ejaculation [81,82]. Additionally, other studies found no correlation between poor semen quality and level of injury, time since injury, or the patient’s age [83,84,85].

The semen of men with SCI is characterized by leukocytospermia, an unusually high concentration of white blood cells [86,87,88]. Leukocytospermia in men with SCI is not attributed to genitourinary tract infection, as antibiotic treatment does not affect sperm quality [80]. Flow cytometric analysis of semen from men with SCI has shown large numbers of activated T- lymphocytes [86]. It is well known that activated T-lymphocytes can exert a damaging effect on other cells through the secretion of cytotoxic cytokines [89,90,91]. A study measuring levels of cytokines in the seminal plasma of men with SCI showed elevated concentrations of interleukin-6 (IL-6), IL-12, IL-1β, tumor necrosis factor alpha (TNF alpha), and interferon-gamma (IFN-γ) in men with SCI compared to healthy age-matched controls [92]. In vitro neutralization of these elevated cytokines improved sperm motility in the semen of men with SCI [93,94].

Zhang et al. sought to determine if activation of the inflammasome complex is a possible mechanism leading to elevated cytokines in the semen of men with SCI. This study reported the presence of the inflammasome component caspase-1 and the adaptor protein ASC in the semen and sperm of men with SCI [95]. Further, a pilot study using oral probenecid to block pannexin-1, a cell wall channel facilitating inflammasome activation, improved sperm motility in men with SCI [96]. These results point to a potential immunologic cause for the abnormal semen quality seen in men after SCI.

## 3. Reproductive Options after SCI

As mentioned, most men with SCI cannot produce an ejaculate during sexual intercourse; thus, a combination of assisted ejaculation and assisted reproductive technology (ART) is warranted. Evaluation begins by determining which ART will be utilized based on safety, laboratory workup, and patient preference. A comprehensive semen analysis is often utilized to aid in deciding between various modalities. The female partner can undergo a gynecological workup to rule out any female factor issues.

### 3.1. Intravaginal Insemination (IVI)

The simplest, minimally invasive, and most cost-effective method of ART is via IVI (termed “at-home insemination”). The total motile sperm count is important for a successful IVI; however, no guidelines exist to define the minimum acceptable number of motile sperm for couples with a male partner with SCI. Before attempting the first intravaginal insemination, the female partner should be familiar with a home ovulation monitoring kit. Multiple studies have demonstrated successful pregnancies and live births via IVI in couples with a male partner with SCI [97,98,99].

### 3.2. Intrauterine Insemination (IUI)

IUI begins by collecting semen from men with SCI via PVS or EEJ under standard safety protocols. The sample is then processed in the laboratory to separate the sperm from the semen. Next, motile sperm are collected and placed inside the woman’s uterus. IUI can be done without stimulating the female partner (unstimulated cycles), or fertility drugs can be prescribed to stimulate the production of eggs and/or ovulation. Contrary to IVI, guidelines concerning the minimum total motile sperm count necessary to attempt an IUI cycle exist and are used to enhance the chances of a successful pregnancy in couples with or without a male partner with SCI [97,100,101].

### 3.3. In Vitro Insemination/Intracytoplasmic Sperm Injection (IVF-ICSI)

If IVI or IUI attempts are unsuccessful in achieving a successful pregnancy, advanced ART is available in the form of IVF-ICSI. In conventional IVF, sperm is retrieved from men with SCI by PVS, EEJ, or surgically extracted from the testis or epididymis. The female partner is stimulated, and the retrieved sperm are then placed in a laboratory dish with the retrieved eggs and placed in an incubator for up to 5 days until fertilization occurs. Embryos that develop to the highest quality blastocyst stage are then placed into the female partner’s uterus. The IVF-ICSI procedure is commonly utilized when the number of motile sperm is too low for the conventional method. In this procedure, a single sperm is injected directly into the egg. Studies have shown lower fertilization rates but similar pregnancy and live birth outcomes in men with SCI compared to men with other causes of male factor infertility. Sperm collected via EEJ and PVS yielded similar IVF/ICSI success rates [102].

## 4. Pursuit of Pleasure and Intimacy after SCI

Sexuality is generally not well addressed in medical, academic, military, and rehabilitation settings. At best, sexual health is limited to issues of sexual dysfunction or impairment, changes in functioning after injury or onset of illness, deteriorating chronic health conditions or disability [103]. However, a more current understanding of disability/ability focuses on well-being, completeness, pleasure, enjoyment, enrichment, and flourishing in life. Such a strength-based model guides the assessment and treatment of sexual health-related issues and conditions as part of the “whole person” [104]. This comprehensive approach reinforces the importance of investigating issues on an individualized basis related to sexual health while incorporating facets of social context, identity, gender, sexual orientation, culture, religion, ethnicity, additional medical and emotional conditions, developmental history, age, beliefs, life experiences, personal choice, diversity of ability and disability, and choice [105]. In fact, the complexity of understanding the individual is the foundation from which evidence-based intervention and support are derived.

Spinal cord injury is a traumatic, life-altering experience usually associated with an initial loss of control across multiple dimensions [106]. Common themes include clinging onto life, reestablishing living, repair and rehabilitation, strategies and adaptation, mourning what is lost, and letting go to allow for what is new and available. The pursuit of pleasure and intimacy follows a similar course in terms of adjustment to maximize individualized defined quality of life.

Sexuality, pleasure, intimacy, gender identities and roles, sexual orientation, eroticism, and reproduction can be defined and given value by the individual. Intimacy follows sharing personal information, feelings, and thoughts that foster being understood, valued, respected, and caring about oneself and each other. Here, the self becomes us, and self-pleasure becomes we as partners-pleasure. The excitement of self-pleasure broadens to the excitement of also pleasing another person. Pleasure brings together the mind and body, thoughts and feelings, knowledge and exploration, the joining of safety-trust-connectedness into awareness, and the opportunity to travel on the road of sexual self-exploration and discovery.

However, for some, there are changes following injury that can produce a challenge to the sexual relationship with oneself and with others. These may include physical appearance, spasms, loss of bowel and bladder function, erectile dysfunction, changes in vaginal lubrication, contractures, weakness, decreased endurance, limited communication, social acceptance, changes in roles, decline in sexual desire, and decrease in verbal/physical expression of affection. Rehabilitation professionals trained in these areas can provide evidence-based and best-practice interventions, along with education in assisting the individual, partner, or couple. Furthermore, educational information must be disseminated regarding safe, respectful sex, including permission and consent, sexually transmitted disease (STD) prevention, and protection against potential violence/abuse. There are also additional healthcare access opportunities to provide screening, education, and support, such as primary care interactions [107].

Individuals with spinal cord injury can also face obstacles, including stereotypes of what constitutes an acceptable relationship, partners, and life activities [108]. Such challenges can impede successful navigation with the intricacy of finding the right person or persons. Misconceptions about spinal cord injury, rooted in a lack of knowledge, misinformation, and limited understanding, can ruin the very foundation for good self-esteem and self-efficacy.

Strong self-efficacy drives a person’s accomplishment and well-being. Strong self-esteem inspires oneself to at least attempt and, most importantly, propels confidence that the person has the right to attempt. However, individuals with low self-efficacy and self-esteem can view challenges such as relationships as threats and avoid or withdraw from them.

Current research informs us that individuals with spinal cord injury can have the same levels of sexual desire as individuals without spinal cord injury. Both groups can have positive sexual relationships, with some having less or more sexual dissatisfaction than others. Personal choice, cultural and religious belief and practice, injury severity, onset of injury, opportunity for intimacy, privacy and access to meet others are examples of factors that can impact the expression of closeness and desire following spinal cord injury.

For the most part, in psychological counseling, when people talk about sex after injury, they are also talking about having a positive and meaningful relationship with another person [108]. When seeking relationships, an opportunity is typically mediated through means such as a friend’s introduction or a random chance meeting. More recently, access to the opportunity to meet has been aided to a degree through online dating services and safety-monitored chat rooms. Still, the challenge for many individuals remains in deciding how much and when to disclose one’s injury to others.

Partner’s satisfaction is crucial in a healthy relationship and most importantly in the case of a partner with SCI. The other partner must understand and accept some of the limitations and obstacles in such a relationship. These include involuntary bladder or bowel accidents during sexual activities as well as position adjustments.

To address the deficit of sex education in healthcare rehabilitation and building or rebuilding sexual intimacy after spinal cord injury requires an educated and open mind, listening rather than prescribing, and a stance of assisting rather than defining or directing. Testimonials offer insight into the life journey: “The best sex comes from open communication and the willingness to be silly, to forget about the inconveniences the disability creates” [109], and “The secret to great sex lies more in the state of your mind than in the state of your body, in the feeling in your heart more than in the feeling in your genitals, and in the quality of the connection more than in the quality of the erection” [110].

## 5. Conclusions

The road toward physical, mental, and emotional rehabilitation and recovery following SCI is marked by numerous challenges and can look very different for each person. To date, multiple safe and effective treatments are available to address many of the sexual and reproductive obstacles posed by SCI. Safe and effective methods of sperm retrieval maximize the changes of biological fatherhood in men with SCI. Applying multiple pharmacological treatment modalities for sexual dysfunction strengthens relationships. However, it is important to reemphasize that this recovery is characterized not just by understanding what remains but also by realizing what can be gained. The pursuit of meaning, fulfillment, and happiness forms the strength of an enriching life for those individuals impacted by SCI.

## 6. Resources/Support

### 6.1. Book Readings

-Regain That Feeling: Secrets to Sexual Self-Discovery: People Living with Spinal Cord Injuries Share Profound Insights into Sex, Pleasure, Relationships, Orgasm, and the Importance of Connectedness—Mitchell S. Tepper, Ph.D., MPH [110].-Enabling Romance: A Guide to Love, Sex and Relationships for People with Disabilities (and the People who Care About Them)—Ken Kroll and Erica Levy Klein [109].

### 6.2. Educational Materials

-PleasureABLE: Sexual Device Manual for Persons with Disabilities [111].-Sex, Love and Intimacy after Spinal Cord Injury [112].

### 6.3. Organizations

-The Miami Project to Cure Paralysis [113].-United Spinal Association [114].-Christopher & Dana Reeve Foundation [115].

## Figures and Tables

**Figure 1 jpm-12-01985-f001:**
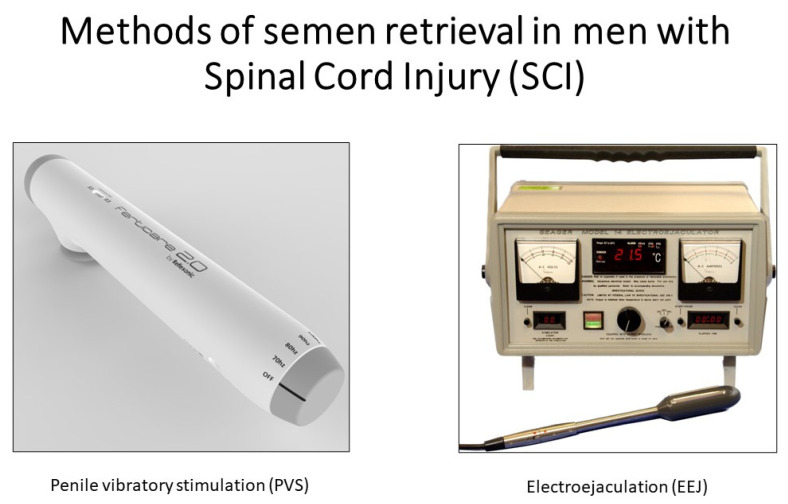
Ferticare 2.0 and Seager Electroejaculation machine are used for semen retrieval in men with SCI.

## Data Availability

Not applicable.

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
