# Peer review of "Sexuality, Intimacy, and Reproductive Health after Spinal Cord Injury"

_jpm, 2022, doi:10.3390/jpm12121985_

Round 1
Reviewer 1 Report
The paper Sexuality, Intimacy, and Reproductive Health after Spinal Cord 2 Injury is written very well, written in a clear and easy to read way. Authors describes sexual dysfunction in SCI patients in an exhaustive way. However, the authors should emphasize that a successful sexual relationship may be disturbed by disorders of the bladder and bowel function may cause involuntary urination and stool during sexual intercourse which may cause fear of sexual intimacy. Therefore, it should be emphasized that in a successful sexual relationship of people with SCI, the partner's acceptance is very important.Author Response
Thank you for highlighting this important issue. A new paragraph was added to the manuscript. Please see lines 301 – 304.
Reviewer 2 Report
The authors present a remarkable review of spinal cord injury sexual dysfunction. The work is well written and easily understood even by those not specifically concerned with this topic. I suggest some minor corrections, however:
- In the abstract I would insert more technical and less "journalistic" language regarding the content of the paper;
- Is this a comprehensive review? or has a systematic database search been done? This is important to include in order to provide an adequate basis for those who want to explore this topic further in a subsequent study.
- Direct damage at the spinal cord level with the sexual component can be determined not only by "direct trauma" after an accident, but also by other conditions that result in compression on the cord and may be equally common or subtle (an example has also been reported for thoracic disc herniations that is often underestimated Armocida D, D'Angelo L, Paglia F, Pedace F, De Giacomo T, Valentino Berra L, Frati A, Santoro A. Surgical management of giant calcified thoracic disc herniation and the role of neuromonitoring. The outcome of large mono centric series. J Clin Neurosci. 2022 Jun;100:37-45. doi: 10.1016/j.jocn.2022.03.046. Epub 2022 Apr 4. PMID: 35390556.)
- The conclusions appear unspeculative and again with a somewhat too "journalistic" slant.
Minor revision
Author Response
- In the abstract I would insert more technical and less "journalistic" language regarding the content of the paper;
- Language was changed with technical terms.
- Is this a comprehensive review? or has a systematic database search been done? This is important to include in order to provide an adequate basis for those who want to explore this topic further in a subsequent study.
- This is a comprehensive review; no systematic database search was performed.
- Direct damage at the spinal cord level with the sexual component can be determined not only by "direct trauma" after an accident, but also by other conditions that result in compression on the cord and may be equally common or subtle (an example has also been reported for thoracic disc herniations that is often underestimated Armocida D, D'Angelo L, Paglia F, Pedace F, De Giacomo T, Valentino Berra L, Frati A, Santoro A. Surgical management of giant calcified thoracic disc herniation and the role of neuromonitoring. The outcome of large mono centric series. J Clin Neurosci. 2022 Jun;100:37-45. doi: 10.1016/j.jocn.2022.03.046. Epub 2022 Apr 4. PMID: 35390556.)
- Non-traumatic etiology was included in the manuscript.
- The conclusions appear unspeculative and again with a somewhat too "journalistic" slant.
- The conclusion was modified according to the important reviewer’s comments and suggestions.